ecology/cognition/behaviour

gillnet, bycatch, seabirds, deterrent, mitigation measures, marine conservation

**Author for correspondence:**
Yann Rouxel
e-mail: yann.rouxel@rspb.org.uk

# Buoys with looming eyes deter seaducks and could potentially reduce seabird bycatch in gillnets

Yann Rouxel[1], Rory Crawford[1], Ian R. Cleasby[2], Pete Kibel[3], Ellie Owen[2], Veljo Volke[4], Alexandra K. Schnell[5] and Steffen Oppel[2]

[1]BirdLife International Marine Programme, c/o the Royal Society for the Protection of Birds Scotland, 10 Park Quadrant, Glasgow, UK
[2]RSPB Centre for Conservation Science, Royal Society for the Protection of Birds, The Lodge, Sandy, UK
[3]Fishtek Marine, Webbers Way, Dartington, Devon, UK
[4]Estonian Ornithological Society, Veski 4, Tartu, Estonia
[5]Department of Psychology, University of Cambridge, Cambridge, UK

YR, 0000-0002-0619-6580; IRC, 0000-0002-4443-0008;
EO, 0000-0003-2073-2420; AKS, 0000-0001-9223-0724;
SO, 0000-0002-8220-3789

Bycatch of seabirds in gillnet fisheries is a global conservation issue with an estimated 400 000 seabirds killed each year. To date, no underwater deterrents trialled have consistently reduced seabird bycatch across operational fisheries. Using a combination of insights from land-based strategies, seabirds' diving behaviours and their cognitive abilities, we developed a floating device exploring the effect of large eyespots and looming movement to prevent vulnerable seabirds from diving into gillnets. Here, we tested whether this novel above-water device called 'Looming eyes buoy' (LEB) would consistently deter vulnerable seaducks from a focal area. We counted the number of birds present in areas with and without LEBs in a controlled experimental setting. We show that long-tailed duck *Clangula hyemalis* abundance declined by approximately 20–30% within a 50 m radius of the LEB and that the presence of LEBs was the most important variable explaining this decline. We found no evidence for a memory effect on long-tailed ducks but found some habituation to the LEB within the time frame of the project (62 days). While further research is needed, our preliminary trials indicate that above-water visual devices could potentially contribute to reduce seabird bycatch if appropriately deployed in coordination with other management measures.

# 1. Introduction

The accidental capture of organisms in fisheries, known as bycatch, has been identified as a primary driver of population declines in several species of marine megafauna [1]. Bycatch is widely recognized as one of the top three threats affecting seabird species globally, both in the number of species affected, and the total number of birds potentially impacted [2]. Hundreds of thousands of seabirds are estimated to be killed in global fisheries each year, with the main sources of mortality coming from gillnet, longline and trawl fisheries [3–5]. Various technical mitigation measures have been developed and have proven effective in reducing seabird bycatch in longline and trawl fisheries [6–8]. However, to date, a ubiquitously effective solution has not been identified to mitigate bycatch in gillnet fisheries, despite an estimated 400 000 seabirds being by-caught in gillnet fisheries each year [5].

Some studies [9–11] suggest that technical measures, such as LED lights, can reduce seabird bycatch in gillnet fisheries. However, these studies often report bycatch reduction of a few species in specific geographical areas, and the reductions observed in these studies have not been observed in fisheries elsewhere (with different species and geographies) [12]. The current best practice for minimizing seabird bycatch is to spatially or temporally exclude gillnet fishing from specific areas or at times when susceptible species are known to aggregate [5]. However, such management measures are highly unpopular among fishers as they reduce fishing income and are only effective if seabird distribution data are adequate to predict seabird distributions in space and time, with potentially unexpected and negative outcomes when such data are not available [13,14]. Therefore, a universally effective method to reduce bycatch in gillnets without the outright banning of gillnets is urgently required.

The Baltic Sea has been identified as a global 'hotspot' for seabird bycatch in gillnet fisheries, with mortalities estimated at 76 000 birds caught annually [5]. The Baltic is particularly important for wintering sea ducks, including long-tailed ducks *Clangula hyemalis* and velvet scoters *Melanitta fusca* [15,16] but is also extensively used by a large fleet of commercial and artisanal gillnet fishermen. The selection of the same productive waters of the Baltic Sea by both fishers and seabirds explains the high level of bycatch [17].

Previous attempts to reduce seabird bycatch in the Baltic Sea by making gillnets more visible underwater did not reduce bycatch. During trials, whereby visual alerts were attached to standard gillnets (e.g. high-contrast panels and LED lights), bycatch remained the same or even increased for long-tailed ducks when white flashing LED lights were used as a deterrent [12,18]. The Baltic Sea has very turbid water with a high sediment load, and even marine-adapted birds may have reduced visual capacity in these waters [19]. A review of seabird cognition suggested that submarine visual signals may be entirely ineffective [20]. Rather than trying to prevent them from becoming entangled once underwater, visually deterring seaducks from diving in the vicinity of gillnets using surface visual signals has been suggested as a potentially more effective approach [20]. We, thus, developed a prototype device, which sits on the ocean's surface, aiming to deter seabirds from diving near and into gillnets.

Escape or fear responses to looming stimuli have been observed in many taxa, ranging from invertebrates and amphibians to primates and birds [21–23], and have been found to trigger a collision-risk signal in avian brains [24,25]. Conspicuous eyespots are more likely to evoke an aversive response in avian species than other stimuli [26–30]. Additional features that enhance behavioural responses from birds include a crescent-shaped reflection inside the pupil, which amplifies the illusion of a spherical eyeball [28], as well as a pupil-to-eye-ratio that was most effective in inducing tonic immobility in chickens (i.e. a natural state of paralysis) [31]. Moreover, a looming eye stimulus displayed on LED screens has been shown to be effective in deterring birds of prey and corvids from airports without signs of habituation [32]. We, therefore, included these features in our prototype device to reduce habituation and enhance deterrence from seabirds in high-risk bycatch zones.

Here, we describe the design of a buoy with a rotating set of looming black-and-white circles, which superficially resemble the staring eyes of a predator. We used these buoys in a controlled experiment in a natural area frequented by seaducks in the Estonian Baltic Sea and counted the number of birds present in areas with and without the experimental buoys. We tested whether the presence of these buoys successfully reduced the number of seaducks and other seabirds within a confined area surrounding these buoys.

# 2. Methods

## 2.1. Looming eyes buoy design

To ensure that the buoy would elicit a strong evasive response by seabirds, we chose a looming eye design with a black and white pattern to offer best contrast and visibility at distance [19]. We designed a 150 mm wide target with a centre 'pupil' of 75 mm in diameter. These measurements are based on the visual acuity of Canada geese *Branta canadensis*, which have one of the lowest acuities measured in birds. We predicted that this design could be detected by a duck at a distance of 80 m during daylight, and at approximately 40 m during twilight (G. Martin 2020, personal communication). The looming eye panels were designed to be 200 mm wide as a result, to ensure that they could be detected from a distance of at least 50 m even during relatively low light levels.

This looming eye design was incorporated into a three-dimensional rotating device consisting of two panels simulating an eye pattern (figure 1). The opposite face of each panel exhibited an eye pattern of different size, in order to create the 'looming' effect when the panels rotate. Panels were assembled and shaped similar to a sinusoidal wind turbine in order to facilitate movement by the wind. Natural wind gusts induced unpredictable movements and speed rotations, which further intensify the likelihood of birds' behavioural responses and reduce chances of habituation [20,33].

The looming eyes buoy (LEB) prototype did not contain electronic or other components vulnerable to corrosion, because in the intended application the device needs to withstand the marine environment with minimum maintenance requirements. Easy handling and low cost are essential for any potential bycatch mitigation device to be effectively incorporated into fishing practices and eventually adopted by fishers [34]. The rotating panels were assembled on an aluminium pole fixed to a standard multi-purpose buoy widely used by the fishing industry for marking gear location. A counterweight was fixed at the base of the pole to increase at-sea balance of the device.

## 2.2. Calculation of potential deterrence range

The prototype LEB that we developed aims to deter birds from gillnets above the water surface, hence it is important to understand the potential effective radius of the LEB. To approximate this radius, we assumed that the horizontal distance that diving seabirds cover during a typical foraging dive would provide the maximum radius over which the LEB would need to elicit a response to act as an effective bycatch mitigation device. We, therefore, examined dive data of species susceptible to gillnet bycatch obtained from individuals tracked with GPS loggers and time-depth recorders at North Atlantic colonies [35,36]. These data revealed that proficient pursuit divers like common guillemots (*Uria aalge*), razorbills (*Alca torda*) and European shags (*Phalacrocorax aristotelis*) rarely travel more than 50 m horizontally in a single dive (table 1; electronic supplementary material, figure S1). While similar dive data are lacking for long-tailed ducks, the foraging behaviour of long-tailed ducks substantially differs from these pursuit divers, because they are benthic feeders and conduct dives of a primarily vertical shape [37,38]. In the absence of proper tracking data, we roughly extrapolated that long-tailed ducks may travel up to 30 metres horizontally per dive (R. Zydelis 2020, personal communication). We, therefore, assumed that limiting dives within a 50 m radius around gillnets could reduce the risk of bird entanglements across a broad range of species susceptible to gillnet bycatch.

## 2.3. Field test methodology

To test whether the LEB had a deterrence effect on seabirds vulnerable to gillnet bycatch, we deployed LEBs in Küdema Bay on the northern coast of the island of Saaremaa (Estonia), a site recognized as a marine Important Bird Area, most notably for wintering and staging waterbird species [39] (figure 2a). Consistent with our assessment of the typical horizontal dive range and the visual acuity of waterbirds, we examined whether the deployment of LEBs would significantly reduce the number of birds in a 50 m radius around the LEB.

Three LEBs were aligned with an oblique angle from the coastline and spaced 100 m apart in order to simulate the presence of a 200 m long gillnet. Smaller buoys were deployed at a 50 m distance from LEBs, creating an elongated hexagon of roughly 25 000 m$^2$ that served as the observation plot for counting birds (figure 2b,c). An identical control plot was deployed at similar distance from the coast, and within 500 m from the treatment plot to ensure that feeding and observation conditions

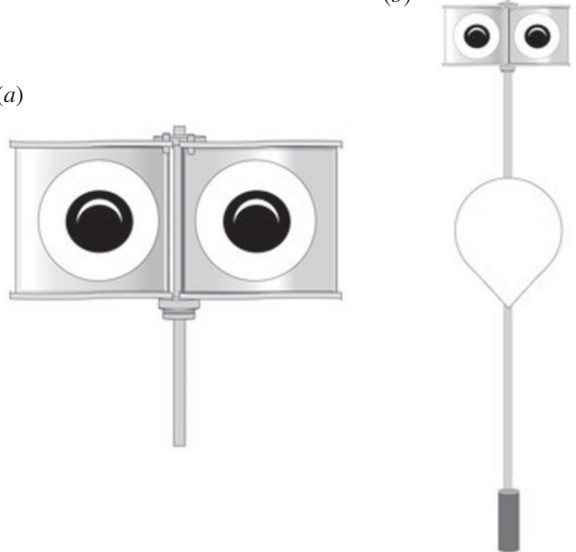

**Figure 1.** (*a*) Looming eyes buoy (LEB) rotating unit with panels and eyes pattern; (*b*) unit fully assembled on pole buoy with counterweight.

**Table 1.** Summary of horizontal distance (m) travelled during a single foraging dive for three diving seabird species derived from tracking data of *N* individuals that were equipped with GPS and time-depth recorders. COMU = *Uria aalge*, RAZO = *Alca torda*, SHAG = *Phalacrocorax aristotelis*.

|  | *N* birds | *N* dives | 1st quartile | median | mean | 3rd quartile | max |
|---|---|---|---|---|---|---|---|
| COMU | 48 | 19 209 | 8.88 | 18.56 | 26.56 | 36.27 | 286 |
| RAZO | 82 | 24 989 | 6.72 | 12.59 | 18.35 | 22.66 | 399 |
| SHAG | 20 | 6224 | 6.29 | 12.45 | 20.14 | 23.36 | 651 |

were similar to the treatment plot. The outline of the control plot was marked with identical small buoys as the treatment plot, and instead of LEBs contained three standard flag buoys, which are used by local fishers to mark gillnets, and are therefore a ubiquitous object type in this area of the Baltic Sea (figure 2*c*). Water depths ranged from 4 to 10 m under each plot (from the buoy closest to shore to the furthest).

Between 6 February and 11 April 2020, seabird abundance (number of individuals per species) was recorded daily within each observation plot every 10 min for 4 h. Birds were identified and counted by a land-based observer (approx. 500 m from plots, figure 2*b*) using a 20–60 × 80 spotting scope and 10 × 45 binoculars. To ensure a good view over the study area, the observation point was located on an elevated part of the shore, approximately 8 m above sea level. We recorded data on a handheld device using an electronic spreadsheet application. Environmental conditions such as temperature, wind speed and direction were recorded using data from the weather station situated in the harbour of Saaremaa, roughly 1 km north of the observation point (figure 2*a*).

We also recorded sea state (Beaufort scale), cloud cover (on a scale from 0 to 8), and visibility (in three categories, 'bad', 'reduced' and 'optimal') for each observation period, because these conditions can influence the number of swimming seabirds observed.

We structured our data collection in a before-after-control-impact design [40] to reduce potential bias due to the random selection of only a single plot for LEB deployment [41]. We switched the location of the LEB and the control (flag) buoys between the two plots in mid-March, resulting in two six-week periods with alternating LEB deployment. Each experimental six-week period included (i) counts 5 days prior to deploying devices to establish a baseline number of seabirds in each plot (pre-phase), (ii) counts for 20 days while LEBs and control buoys were deployed to test for aversive effects as well as habituation effects (experimental phase), and (iii) counts for 5 days after LEB and control buoy removal to test for a memory effect (post-phase).

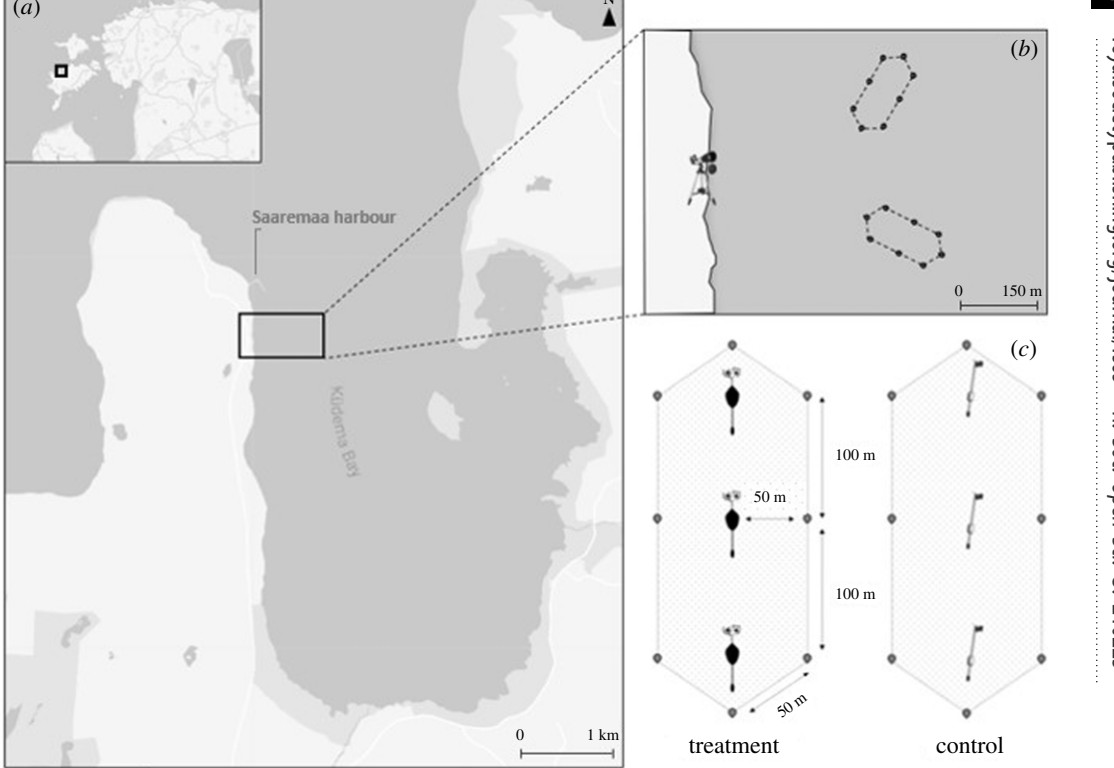

**Figure 2.** (*a*) Localization of fieldwork in Küdema Bay, Saaremaa Island, Estonia; (*b*) close-up map of the observation point and experimental plots; (*c*) schematic of the delineation of treatment and control plots in which seabirds were counted.

## 2.4. Statistical analysis

We first explored whether the presence of LEBs can account for any variation in the counts of seabirds, and then tested specifically for an effect of LEB deployment in a before-after-control-impact (BACI) design analysis—a similar approach was adopted by Jiménez *et al.* [42] in their study assessing seabird bycatch reduction in pelagic longline fleets. Counts of seabirds in a small area are subject to large temporal variation due to diurnal and seasonal movements, temporal disturbances, the influence of tidal currents, wind/weather factors, and other aspects that are difficult to measure [43–45]. Due to the large number of potential environmental factors that may affect seabird abundance and the fact that some of these variables may be correlated, we first conducted an exploratory analysis using an analytical approach that can accurately identify the relative importance of variables under these conditions [46–48]. To assess whether the presence of the LEBs had any measurable effect on the instantaneous counts of long-tailed ducks, we used a random forest algorithm to relate the number of seabirds counted in a given 10 min interval to 12 variables that could plausibly explain the number of counted birds: day of the year, time of day, weather, temperature, wind speed and direction, sea state, cloud cover, visibility and the identity of the observer. In addition to these environmental noise variables, we also included the two experimental variables, namely the presence of LEBs and the experimental phase (before LEB deployment, during and after LEB deployment).

A random forest is a machine learning algorithm based on ensembles of regression trees that can accommodate a large number of predictor variables and yields highly accurate predictions [46,47,49]. To assess which variables had the greatest influence on instantaneous long-tailed duck abundance, we used a permutation procedure that assesses the increase in mean squared error of the random forest model after randomly permuting a given variable [50–52]. We implemented this assessment using the function 'importance' in the R package 'randomForest' [53] with 100 permutations per variable. We present results as relative variable importance, with the most important variable assigned a value of 100%. We also evaluated the explanatory power of the random forest model by calculating the correlation coefficient between observed abundances and those predicted from the model ($r_s$).

Since a random forest is a non-parametric algorithm, the direction and size of effects by given variables cannot be expressed with numeric parameter estimates. To estimate the effect size of the LEB, we also used a classical BACI analysis to specifically test whether, and quantify by how much, the presence of LEBs affected the abundance of long-tailed ducks. For this analysis, we used a generalized linear mixed model (GLMM) that tested for an interaction between the presence of the LEB and the phase of the experiment (before deployment or during and after deployment), thus overcoming any potentially confounding effects that may have occurred if LEBs had been placed in an area that was naturally less attractive to long-tailed ducks [41]. We fit two corresponding models, with and without the interaction of interest, and used a likelihood-ratio test to assess the statistical significance of the interaction [54]. Each model contained the most important variables identified by the exploratory random forest model (day, time of day, observer, presence of LEB, and project phase, see Results) as fixed factors, and accounted for non-independence of serial observations during the same project phase by including a random intercept for each project phase. We fitted these models with the function 'glmmTMB' in the R package 'glmmTMB' using a negative binomial error distribution [55].

Two additional effects of the LEB are plausible and relevant for its potential value as deterrence device: seaducks may memorize the presence of a 'predator' and vacate an area even after the LEB has been removed, or seaducks may become accustomed to the device and the deterrence effect may diminish over time. While the memory effect could potentially displace seaducks from large areas of suitable habitat and thus have negative consequences on their energy budgets, the habituation effect could undermine the effectiveness of the LEB as a deterrence device.

We tested the memory effect specifically with a similar GLMM as described above but removed the baseline data before LEB deployment. We considered the phase during LEB deployment as the baseline and compared long-tailed duck abundance after LEBs had been removed to the number during LEB presence by including a treatment × phase interaction. To test the habituation effect, we introduced a continuous covariate indicating the number of days that had elapsed since LEB deployment (0–25) and fitted a similar GLMM testing for an interaction between the day since deployment and the presence of an LEB.

For each of the three pairs of models (overall effect, memory and habituation), we report the overall significance of the model based on the likelihood-ratio test, the mean estimates and standard errors of the relevant interaction parameter, and the predicted effect sizes calculated as the number of long-tailed ducks within 50 m of the LEB compared with the number in the respective control plot during the same experimental phase.

## 3. Results

We counted seabirds on 62 days during 250 h of observation and recorded a total of 11 118 seabirds of 18 different species within our two experimental plots. The vast majority of reported birds were long-tailed ducks (91.4%), while the second most recorded species, eiders, represented 2.3% (table 2). Given the limited number of other species, we focused our statistical analysis on long-tailed ducks only. On average, 3.5 long-tailed ducks were present in each of our two experimental plots during the 2941 individual counts, with a range of 0 (15% of counts) to 120 long-tailed ducks. Abundance varied over time, with a mean of 2.3 long-tailed ducks in February ($n = 1011$ counts), 4.9 in March ($n = 1404$) and 1.9 in April ($n = 526$).

The random forest model examining the influence of environmental variables on long-tailed duck abundance explained 32% of the variation in the data and counts predicted from the model were correlated positively with observed counts ($r_s = 0.564$). The presence of the LEB was the most important variable explaining long-tailed duck abundance, followed by observer and the time of day (figure 3).

The parametric BACI analysis indicated that there was a statistically significant interaction between the presence of LEBs and the experimental phase, indicating that long-tailed duck abundance declined at times and in places where LEBs were present, but not at other times of the experiment when no LEBs were present (LR-test, $\chi^2 = 5.43$, $p = 0.019$). The parameter estimates for the treatment × phase interaction effect was 0.558 (s.e.: 0.239, $p = 0.019$). This model predicted that the mean number of long-tailed ducks decreased by 23–24.9% during the presence of LEBs but was 5.8% higher in an area before LEBs were installed compared with the control area where no LEBs were installed (figure 4). Because this model also predicted that long-tailed duck abundance would be reduced by 45.4–53%

**Table 2.** Total number of seabirds per species counted within two experimental plots (figure 2) between February and April 2020 in Küdema Bay, Estonia. The 'treatment' plot contained three LEBs, while the 'control' plot contained three standard flag buoys, and control and treatment were switched between phase 1 and 2 of the project.

| | long-tailed duck | | eiders[a] | | mergansers[a] | | gulls[a] | | others[b] | |
|---|---|---|---|---|---|---|---|---|---|---|
| | control | treatment | control | treatment | control | treatment | control | treatment | control | treatment |
| pre-phase 1 | 236 | 358 | 11 | 151 | 12 | 40 | 11 | 10 | 19 | 51 |
| phase 1 | 2550 | 1221 | 86 | 1 | 17 | 25 | 71 | 16 | 31 | 23 |
| post-phase 1 | 1241 | 686 | 8 | 0 | 12 | 24 | 9 | 7 | 9 | 10 |
| phase 2 | 1552 | 2176 | 0 | 0 | 26 | 18 | 103 | 31 | 17 | 10 |
| post-phase 2 | 145 | 2 | 0 | 0 | 51 | 19 | 10 | 3 | 9 | 0 |
| total | 5724 | 4443 | 105 | 152 | 118 | 126 | 204 | 67 | 85 | 94 |

[a]Species were grouped to simplify representation in this table (e.g. eiders include Steller's and common eiders).
[b]others include = scoters, Grebes, alcids, cormorants, other diving seaducks and unidentified species.

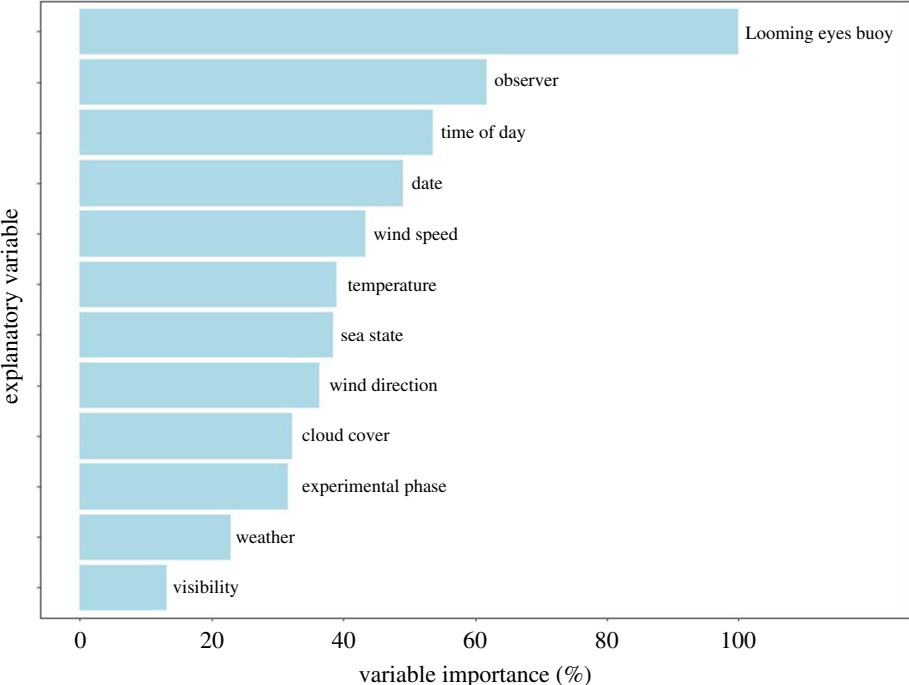

**Figure 3.** Relative importance of 12 predictor variables influencing counts of long-tailed ducks in plots extending 50 m around a string of three Looming eyes buoys (LEBs) and a control plot during different phases of a controlled experiment in Estonia in spring 2020. Variable importance was derived from a permutation procedure quantifying the increase in mean squared error of a random forest model.

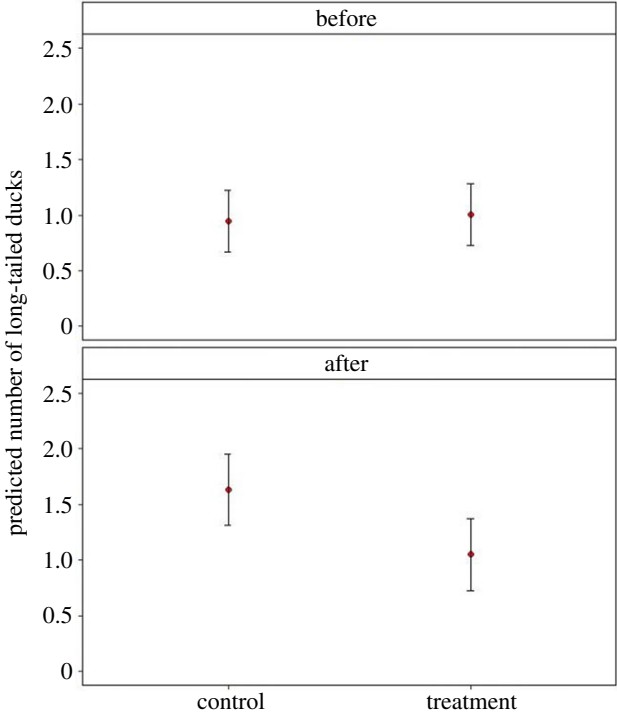

**Figure 4.** Predicted mean abundance (±s.d.) of long-tailed ducks in two plots extending 50 m around Looming eyes buoys (LEBs) during a controlled experiment in Estonia in spring 2020. 'before' indicates the baseline condition before LEBs were placed in the 'treatment' plot, 'after' includes the phases during and after deployment of the LEB in the respective 'treatment' plot, which differed between the first and second experimental phase. Predictions were based on a GLMM with interactions between treatment and experimental phase.

shortly after the LEBs had been removed, we investigated this 'memory' effect in greater detail by removing the baseline data before LEB deployment. This detailed model to assess the memory effect indicated a strong effect of the LEB (LR-test, $\chi^2 = 148.56$, $p < 0.001$) with a strong treatment × before/after interaction ($-5.22 \pm 0.85$, $p < 0.001$). However, this model also revealed contrasting results between the first and the second experimental phase with a very strong effect of the treatment × phase interaction ($4.87 \pm 0.77$, $p < 0.001$): while the number of long-tailed ducks increased after LEB removal from both the treatment and the control plots during the first experimental phase, the opposite was the case during the second experimental phase when numbers of long-tailed ducks decreased in both experimental and control plots after LEB removal. There was no memory effect of the LEB during the first phase, because 35.8% more long-tailed ducks returned to the treatment plot compared with the control plot within 5 days of our post-deployment observations; however, only a pair of long-tailed ducks were observed in the treatment plot after LEB removal following the second phase, and thus 318% fewer than in the control plot at that time (electronic supplementary material, figure S2).

A similar difference between the two experimental phases was observed with respect to habituation. Our habituation model found an overall strong effect of the LEB (LR-test, $\chi^2 = 86.741$, $p < 0.001$), but a very small effect of the treatment × day interaction ($0.039 \pm 0.033$, $p = 0.244$) due to the opposite direction of change in long-tailed duck abundance in the two experimental phases. While the number of long-tailed ducks increased over time in both control and treatment plots during the first experimental phase, numbers declined in both plots over time in the second experimental phase in early April (electronic supplementary material, figure S3). However, the relative rate of change in long-tailed duck abundance over time was on average 6.8 times greater in the treatment plot compared with the control plot in both phases, indicating that some habituation to the LEB occurred within 25 days (electronic supplementary material, figure S3).

## 4. Discussion

Our results suggest that our prototype LEB device can reduce the number of long-tailed ducks by approximately 20–30% within a 50 m radius. If deployed at regular intervals on gillnets or other static gears that pose a mortality risk to this species, bycatch numbers could decrease by a similar order of magnitude. Follow-up testing of this device is needed to confirm its potential in tackling seabird bycatch in commercial fishing conditions, using a paired-trial experiment to compare control and experimental nets [12], ideally over an extended-time period to examine potential habituation effects. Estimated annual bycatch of long-tailed ducks in gillnets in the Baltic is about 22 000 birds [56]. If the effectiveness of LEBs is confirmed, however, and these devices are progressively adopted by local gillnet fishermen, this measure could potentially save several thousands of long-tailed ducks each year in the Baltic sea alone from accidental entanglements.

We found no clear effect that would suggest that areas where LEBs had been deployed would keep deterring ducks after the LEB had been removed, but we found evidence to suggest that ducks get habituated to the LEB and the deterrent effect of the LEB might diminish over time. However, these analyses were potentially confounded by the seasonality of seaduck presence in our study area, which resulted in contrasting patterns between our two experimental phases in late February and April: the number of long-tailed ducks increased between February and March, and then decreased between March and April as many individuals departed on migration. Therefore, fewer birds were present in the general study area during and especially after the second experimental phase, which may have exaggerated the memory effect observed after the second experimental phase. Despite the contrasting phenology, the relative rate of change in long-tailed duck abundance over time during LEB presence was consistently greater in the treatment plot, which suggests that long-tailed ducks get habituated to the LEB and its deterrence effect diminishes. Given that the typical gillnet is only deployed for a maximum of a few days—after which catches quickly deteriorate [57]—the habituation effect during actual gillnet fishing operations may be much smaller than the effect we observed in our extended experiment. However, we strongly recommend that any tests of the LEB with real gillnets be carried out during a time when seaduck abundance is not subject to phenological declines to avoid confounding effects that may mask or accentuate differences due to the presence of the LEB.

Given that the visual stimulus used to develop LEBs can trigger escape or fear responses in numerous bird species and other taxa [21–24,58,59], we suspect that similar results could be observed with other seabird species, in particular with seaducks displaying similar foraging strategies. However, our study area did not have a sufficient number of other species present in our study plots to test the effect of

the LEB on other species. Follow-up testing in areas where the abundance and diversity of seabirds is less dominated by a single species should help determine the potential for LEBs to be a multi-species bycatch mitigation measure. The risk of LEBs having a significant species-specific effect—similar to what has been observed for LED lights on nets [10–12]—cannot be ruled out at this stage.

In our study, we assumed that the area over which birds suffer a high risk to dive into a gillnet extends to a 50 m distance from the net, but we emphasize that no empirical data exist to corroborate that assumption. Available tracking data indicated that pursuit-diving seabirds rarely travel greater than 50 m horizontally in a single dive [35,36]. If birds travel farther than 50 m underwater, bycatch reduction from our LEB device may be lower than we extrapolate, unless the deterrent effect of the LEB also extends further than 50 m. Bird detection of the eye pattern and looming effect might also occur at a larger distance than the 50 m threshold, depending on the size of the rotating eyes mounted on the buoy. There is, however, a practical limit to how large this rotating device can be made while still being transportable by artisanal fishermen.

Given the upper limit of the size of the rotating eyes display, the visual stimulus to seabirds that would elicit vigilance behaviour will probably decrease with distance from the LEB [60]. If LEBs are being perceived by birds as potential predation risk, the probability of avoidance behaviour should increase as predation risk increases closer to the LEB [61]. Because of the behavioural response being more likely closer to the LEB, the deterrent effect of LEBs might have been higher if we had used study plots less than 50 m from LEBs. However, reducing the size of our observation plot to capture a potentially higher deterrence effect would have reduced the number of birds present in observation areas, and thus reduced the statistical power to detect a deterrence effect.

Purposefully triggering escape responses and displacing wildlife from feeding grounds poses ethical questions, even when the deterrence is intended to reduce imminent mortality risk. Two lines of evidence of our experiment suggest that the deleterious effect on long-tailed ducks is probably minimal: (i) we found that ducks probably habituate to the LEBs, and may no longer be afraid of the looming eyes after two to three weeks of deployment at the same site; and (ii) we found that after the first experimental phase, long-tailed duck numbers increased more rapidly in the treatment plot than in the control plot, suggesting that an area is unlikely to be avoided after the LEB has been removed. Although we found total avoidance by long-tailed ducks following the removal of the LEB in the second experimental phase, this was probably a consequence of rapidly declining duck numbers in the entire study area due to the onset of migration, rather than a lasting deterrence effect of the LEB. Thus, the displacement effect of the LEB is probably limited and of short duration. The true extent of this effect should, however, be analysed in depth before this measure is more widely adopted, through progressive use of LEBs in test fisheries, paired with monitoring of bird presence in the fishing area over time.

Special considerations to fishing practices were included during the mitigation device development. While a four-panel device would have probably enhanced the looming effect at close range, we decided instead to develop a two-panel version which offered a better compromise between effect and fishing practicality (e.g. easier to stack on the boat, less prone to damage when handled, etc.). As a strictly above-water measure, fisher's nets do not need to be altered when using the LEB, which greatly reduces the impact on target catches compared with underwater mitigation strategies. The low probability to affect fish catch is a direct advantage of this method, which should increase acceptance from fishers and willingness to deploy this mitigation device during commercial operations. The current LEB prototypes were primarily developed as a 'proof of concept' for tests in controlled conditions. Following these initial promising outcomes, further work is needed to develop a smaller and lighter design of the LEB to ensure fishing practicality with gillnets is not negatively affected by its use.

If confirmed effective at reducing seabird bycatch in gillnet fisheries, LEB devices could offer the industry and fisheries managers a less economically damaging alternative to fishing closures. LEB devices could easily be used in coordination with other bycatch mitigation measures (e.g. alongside pingers to reduce marine mammals bycatch) or management approaches (e.g. fishing depth restrictions, seasonal closures, etc.). Used in coordination, LEBs along with other measures are likely to achieve much greater results in terms of bycatch reduction while keeping socio-economic impacts to a minimum. Further research is therefore needed to understand the limits, risks and effects associated with the usage of LEBs during commercial fishing operations.

Data accessibility. Data available from the Dryad Digital Repository: https://doi.org/10.5061/dryad.12jm63xxh [62].
Authors' contributions. Y.R. conceived the tested floating device, designed the study, coordinated the study and drafted the manuscript; R.C. helped in conceiving the study and draft of the manuscript; I.R.C. carried out data analysis of the

horizontal diving patterns and revised the manuscript; P.K. carried out the design and production of the floating device and revised the manuscript; E.O. collected field data on seabirds' diving patterns; V.V. carried out the fieldwork and data collection of the study and helped in drafting the manuscript; A.K.S. participated in the design of the study, the design of the floating device and revised the manuscript; S.O. carried out the statistical analyses of the study, participated in the design of the study and helped draft the manuscript. All authors gave final approval for publication and agree to be held accountable for the work performed therein.

Competing interests. P.K. is Managing director and co-founder of Fishtek Marine, which manufactured the Looming eyes buoys for this project.

Funding. Funding was received from the National Geographic Society and the Baltic Sea Foundation. A.K.S. was awarded funding from the BBSRC Flexible Talent Mobility Account during conceptualization and was supported by a Royal Society Newton International Fellowship during writing. None of the funders had any role in study design, collection, analysis and interpretation of data, in the writing of the report or in the decision to submit this article to publication.

Acknowledgements. We are particularly grateful for the funding support from the National Geographic Society which allowed us to carry out this work in Estonia, as well as the support from the Baltic Sea Foundation to cover the development and production of the Looming eyes buoys. Sincere thanks are due to our fieldwork team—Andres Kalamees, Andrus Kuus, Mati Martinson, Rein Nellis, Maarja Nõmm, Maris Sepp and Ainar Unus—and the Estonian Environmental Agency, without whom this project would have not been possible. We are also grateful to all fieldworkers who collected data as part of the RSPB FAME/STAR tracking projects, which helped better understanding seabirds' diving patterns.

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
