## [Peer Review File · Royal Society Open Science]

Review History

RSOS-210225.R0 (Original submission)

Review form: Reviewer 1

Is the manuscript scientifically sound in its present form?

Yes

Are the interpretations and conclusions justified by the results?

Yes

Is the language acceptable?

Yes

Do you have any ethical concerns with this paper?

No

Have you any concerns about statistical analyses in this paper?

No

Recommendation?

Accept as is

Comments to the Author(s)

I would like to congratulate the authors on preparing this nice manuscript on a very important topic for seabird conservation. I am probably a bad reviewer, but I could not find anything in the manuscript that I would disagree with or could ask for improvement. Truly great job!

Review form: Reviewer 2**Is the manuscript scientifically sound in its present form?**

Yes

Are the interpretations and conclusions justified by the results?

Yes

Is the language acceptable?

Yes

Do you have any ethical concerns with this paper?

No

Have you any concerns about statistical analyses in this paper?

No

Recommendation?

Accept with minor revision (please list in comments)

Comments to the Author(s)

This paper is a valuable contribution presenting a novel mitigation measure with the potential of reducing bycatch of seabirds in gillnet fisheries. Due to the low number of mitigation measures for this threat, the present manuscript is of great conservation importance. The ms is very well written and structured, easy to read. The MS describes a novel device ("Looming-eye buoy" LEB) with the potential to reduce this bycatch, presenting detailed technical information. The experiment is focused on the "proof of concept" of the LEB. I think the experimental trials were well designed and data analyses are appropriate; therefore, from my point of view the results are robust. . I have provided very few suggestions below.

A potential reduction of about 20-30% in bycatch appears low, however, in the absence of other effective mitigation measures, it seems critical to continue conducting research on this promising device. Two important points are: to confirm the level of effectiveness in a study with fishing nets, since the results assume that the device has an effect in 50m around; and to extend the experiment with fishing net to a longer period to contemplate the potential habituation effect. A year-round study should be a good starting point, but this would depend in the bycatch rates. The authors have mentioned some of this, however I think it should be more explicit in detailing how one could advance in demonstrating the effectiveness of the device in real fishing conditions. In addition, It will also be important to determine in the future, if the same weaknesses that have LED lights (lines52-55) apply or not to LEBs. In fact, the proof of concept is based on a single species.

Although there are not many alternatives, an effort should be made to expand into the Discussion what is mentioned in the last sentence of the summary.

Analyses. Note that a similar approach was used in recent study on the effectiveness of mitigation measures for seabirds (<https://doi.org/10.1016/j.biocon.2020.108642>); the use of a random forest to evaluate possible variables of relevance, including ancillary variables, and then generalized models (GAMM instead GLMM, but similar) to determine the effect size of the variables of interest, considering relevant covariates. Perhaps it would be reasonable to include this reference.

Line 236. Please explain why Negative binomial error distribution was used.

Lines 385-388. The way it is worded, it seems of concern. I suggest mentioning that this effect should be analysed before it is universally adopted, through progressive use of the device. Obviously, the first thing is to demonstrate the efficiency with the fishing nets.

Decision letter (RSOS-210225.R0)

Dear Mr Rouxel

On behalf of the Editors, we are pleased to inform you that your Manuscript RSOS-210225 "Buoys with looming eyes deter seaducks and could potentially reduce seabird bycatch in gillnets" has been accepted for publication in Royal Society Open Science subject to minor revision in accordance with the referees' reports. Please find the referees' comments along with any feedback from the Editors below my signature.

Please submit your revised manuscript and required files (see below) no later than 7 days from today's (ie 29-Mar-2021) date. Note: the ScholarOne system will 'lock' if submission of the revision is attempted 7 or more days after the deadline. If you do not think you will be able to meet this deadline please contact the editorial office immediately.

on behalf of Dr Agustina Gómez-Laich (Associate Editor) and Kevin Padian (Subject Editor)
openscience@royalsociety.org

Associate Editor Comments to Author (Dr Agustina Gómez-Laich):

Dear authors,

The manuscript entitled “Buoys with looming eyes deter seabirds and could potentially reduce seabird bycatch in gillnets” has now been seen by two reviewers, both of whom found the work novel, interesting, well-written and well-structured. However, Reviewer#2 raised some few concerns about how effective the device would be under more realistic conditions. In addition, Reviewer#2 suggest to expand the discussion on how this method could be used with other management measures. Additionally to Reviewer#2 comments and suggestions, I suggest incorporating at least a reference in the paragraph in which the potential distance that birds travel underwater is presented (lines 359-376).

Reviewer comments to Author:

Reviewer: 1

Comments to the Author(s)

I would like to congratulate the authors on preparing this nice manuscript on a very important topic for seabird conservation. I am probably a bad reviewer, but I could not find anything in the manuscript that I would disagree with or could ask for improvement. Truly great job!

Reviewer: 2

Comments to the Author(s)

This paper is a valuable contribution presenting a novel mitigation measure with the potential of reducing bycatch of seabirds in gillnet fisheries. Due to the low number of mitigation measures for this threat, the present manuscript is of great conservation importance. The ms is very well written and structured, easy to read. The MS describes a novel device (“Looming-eye buoy” LEB) with the potential to reduce this bycatch, presenting detailed technical information. The experiment is focused on the “proof of concept” of the LEB. I think the experimental trials were well designed and data analyses are appropriate; therefore, from my point of view the results are robust. . I have provided very few suggestions below.

A potential reduction of about 20-30% in bycatch appears low, however, in the absence of other effective mitigation measures, it seems critical to continue conducting research on this promising device. Two important points are: to confirm the level of effectiveness in a study with fishing nets, since the results assume that the device has an effect in 50m around; and to extend the experiment with fishing net to a longer period to contemplate the potential habituation effect. A year-round study should be a good starting point, but this would depend in the bycatch rates. The authors have mentioned some of this, however I think it should be more explicit in detailing

how one could advance in demonstrating the effectiveness of the device in real fishing conditions. In addition, It will also be important to determine in the future, if the same weaknesses that have LED lights (lines52-55) apply or not to LEBs. In fact, the proof of concept is based on a single species.

Although there are not many alternatives, an effort should be made to expand into the Discussion what is mentioned in the last sentence of the summary.

Analyses. Note that a similar approach was used in recent study on the effectiveness of mitigation measures for seabirds (<https://doi.org/10.1016/j.biocon.2020.108642>); the use of a random forest to evaluate possible variables of relevance, including ancillary variables, and then generalized models (GAMM instead GLMM, but similar) to determine the effect size of the variables of interest, considering relevant covariates. Perhaps it would be reasonable to include this reference.

Line 236. Please explain why Negative binomial error distribution was used.

Lines 385-388. The way it is worded, it seems of concern. I suggest mentioning that this effect should be analysed before it is universally adopted, through progressive use of the device. Obviously, the first thing is to demonstrate the efficiency with the fishing nets.

===PREPARING YOUR MANUSCRIPT===

Your revised paper should include the changes requested by the referees and Editors of your manuscript. You should provide two versions of this manuscript and both versions must be provided in an editable format:
 one version identifying all the changes that have been made (for instance, in coloured highlight, in bold text, or tracked changes);
 a 'clean' version of the new manuscript that incorporates the changes made, but does not highlight them. This version will be used for typesetting.
 Please ensure that any equations included in the paper are editable text and not embedded images.

===PREPARING YOUR REVISION IN SCHOLARONE===

Author's Response to Decision Letter for (RSOS-210225.R0)

See Appendix A.

Decision letter (RSOS-210225.R1)

Dear Mr Rouxel,

I am pleased to inform you that your manuscript entitled "Buoys with looming eyes deter seaducks and could potentially reduce seabird bycatch in gillnets" is now accepted for publication in Royal Society Open Science.

on behalf of Dr Agustina Gómez-Laich (Associate Editor) and Kevin Padian (Subject Editor)

Appendix A

Dear Editors,

Thank you very much for accepting that our Manuscript RSOS-210225, is to be published in Royal Society Open Science subject to minor revisions. We have listed below – in a table format – the point-by-point responses to reviewers comments. These edits/comments can also be found in the “tracked final manuscript” (with Tracked Changes). We hope that those changes will meet the reviewers recommendations. Regarding the general comment “*few concerns about how effective the device would be under more realistic conditions*”; we very much agree that further testing – in real fishing conditions– is absolutely needed prior to confirm the potential of this measure as a bycatch mitigation device. This manuscript focuses on the “proof of concept” and through a bird presence approach (behavioural analysis), rather than through fishery-based trials. Current results only suggest what might be the device’s potential to reduce seabird bycatch with gillnets, but with several uncertainties needed to be answered. We hope that the discussion section of this manuscript reflects this. We will also be conducting follow-up testing of this device – in a commercial gillnet fishery setting this time – in 2022, prior to any official conclusion.

Response to editors/reviewers:

Editors/reviewers comment	Response
Analyses. Note that a similar approach was used in recent study on the effectiveness of mitigation measures for seabirds (https://doi.org/10.1016/j.biocon.2020.108642); the use of a random forest to evaluate possible variables of relevance, including ancillary variables, and then generalized models (GAMM instead GLMM, but similar) to determine the effect size of the variables of interest	Added at lines 198/199 “- a similar approach adopted by Jiménez et al. (2020) in their study assessing seabird bycatch reduction in pelagic longline fleets”. Added in the references list:  • Jiménez, S., Domingo, A., Winker, H., Parker, D., Gianuca, D., Neves, T., ... & Kerwath, S. (2020). Towards mitigation of seabird bycatch: Large-scale effectiveness of night setting and Tori lines across multiple pelagic longline fleets. Biological Conservation, 247, 108642.
Line 236. Please explain why Negative binomial error distribution was used	The use of Negative binomial error distribution with ‘glmmTMB’ is explained by the zero-inflated - containing more zeros than would be expected from the standard error distributions used in GLMMs - and overdispersed nature of the data — with the variance being larger than the

	mean. The negative binomial distribution is also recognised often as the best fit for abundance data (Warton 2005, Brooks et al. 2017).  • Brooks, M. E., Kristensen, K., van Benthem, K. J., Magnusson, A., Berg, C. W., Nielsen, A., ... & Bolker, B. M. (2017). Modeling zero-inflated count data with glmmTMB. BioRxiv, 132753. • Warton, D. I. (2005). Many zeros does not mean zero inflation: comparing the goodness-of-fit of parametric models to multivariate abundance data. Environmetrics, 16, 275–289.
“[...] I think it should be more explicit in detailing how one could advance in demonstrating the effectiveness of the device in real fishing conditions. “	Amended at lines 330/335 “If deployed at regular intervals on gillnets or other static gears that pose a mortality risk to this species, bycatch numbers could decrease by a similar order of magnitude. Follow-up testing of this device is needed to confirm its potential in tackling seabird bycatch in commercial fishing conditions, using a paired-trial experiment to compare control and experimental nets (Field et al. 2019), ideally over an extended -time period to examine potential habituation effects.”
“[...] to determine in the future, if the same weaknesses that have LED lights (lines52-55) apply or not to LEBs. In fact, the proof of concept is based on a single species”	Added at lines 363/367 “Follow-up testing in areas where the abundance and diversity of seabirds is less dominated by a single species should help determine the potential for LEBs to be a multi-species bycatch mitigation measure. The risk of LEBs having a significant species-specific effect – similar to what has been observed for LED lights on nets (Mangel et al. 2018, Field et al. 2019, Bielli et al. 2020) – cannot be ruled out at this stage”.
“[...] I suggest incorporating at least a reference in the paragraph in which the potential distance that birds travel underwater is presented (lines 359-376).”	Added references Wakefield et al. 2017; Browning et al. 2018 Note that horizontal dive patterns were analysed only for this study (see Calculation of potential deterrence range l.219 of this manuscript), based on tracking data also used in referenced studies, but which did not explore that aspect of diving behaviour. As a result – and to the best of our knowledge – no other studies have explored horizontal dive patterns for birds and no references are available besides our calculation.

“Lines 385-388. The way it is worded, it seems of concern. I suggest mentioning that this effect should be analysed before it is universally adopted, through progressive use of the device. Obviously, the first thing is to demonstrate the efficiency with the fishing nets.”	Added/amended at Lines 394/398: “Thus, the displacement effect of the LEB is likely limited and of short duration. The true extent of this effect should, however, be analysed in depth before this measure is more widely adopted, through progressive use of LEBs in test fisheries, paired with monitoring of bird presence in the fishing area over time”.
“an effort should be made to expand into the Discussion what is mentioned in the last sentence of the summary →[...] if appropriately deployed in coordination with other management measures”	Added/amended in the manuscript last paragraph. “If confirmed effective at reducing seabird bycatch in gillnet fisheries, LEB devices could offer the industry and fisheries managers a less economically damaging alternative to fishing closures. LEB devices could easily be used in coordination with other bycatch mitigation measures (e.g. alongside pingers to reduce marine mammals bycatch) or management approaches (e.g. fishing depth restrictions, seasonal closures, etc.). Used in coordination, LEBs along other measures are likely to achieve much greater results in terms of bycatch reduction while keeping socio-economic impacts to a minimum. Further research is therefore needed to understand the limits, risks and effects associated with the usage of LEBs during commercial fishing operations”.